# Body Image in Patients with Marfan Syndrome

**DOI:** 10.3390/jcm9041015

**Published:** 2020-04-03

**Authors:** Laura Birke Hansen, Yskert von Kodolitsch, Friedrich Schroeder, Dieter Benninghoven

**Affiliations:** 1Muehlenberg-Clinic for Rehabilitation, 23714 Bad Malente-Gremsmühlen, Germany; laurabirke@gmail.com (L.B.H.); friedrich.schroeder@drv-nord.de (F.S.); 2Clinic of Cardiology at the University Heart Centre, University of Hamburg, 20246 Hamburg, Germany; kodolitsch@uke.de; 3Clinic of Psychosomatic Medicine and Psychotherapy, University of Luebeck, 23538 Luebeck, Germany

**Keywords:** Marfan syndrome, body image

## Abstract

This study aimed to assess body image impairments of individuals with Marfan syndrome and to determine to what extent psychological, physical, and sociodemographic factors influence body image. We assessed the physical fitness and psychosocial health of 42 patients with Marfan syndrome at the beginning of an inpatient rehabilitation program. All participants filled out a body image questionnaire consisting of two scales: (1) Negative Body Evaluation and (2) Vital Body Dynamics. We compared body image data of the study sample with the German representative norming sample and carried out two separate regression analyses in order to determine which variables were associated with the two dimensions of patients’ body image. Body image of individuals with Marfan syndrome appeared to be significantly impaired, with high percentile ranks for Negative Body Evaluation (women = 88, men = 91) and low percentile ranks for Vital Body Dynamics (women = 11, men = 4). Vital Body Dynamics was predicted by age (*p* = 0.016) and by depression (*p* < 0.001), and Negative Body Evaluation was predicted by anxiety (*p* = 0.001). Body image in individuals with Marfan syndrome is not primarily determined by objective measures of fitness or by objective cardiac impairment but by psychological variables like depression and anxiety and by age. This finding can inform treatment and rehabilitation concepts. Accepting Marfan syndrome, including the acceptance of being visually different, may not only demand medical treatment and physical rehabilitation but also psychological treatment for depression and anxiety.

## 1. Background

Marfan syndrome is an autosomal dominant genetic disorder of the connective tissue. Clinical manifestations are variable and affect different organ systems. The most severe problems from a medical perspective are defects of the aorta and of the heart valves. Beyond cardiovascular manifestations, patients with Marfan syndrome also have to cope with reduced visual acuity and even blindness due to lens dislocation. Consequently, many patients wear eyeglasses. Obesity is often a problem in older patients [1]. Skeletal abnormalities may include an elongated face, torso, and limbs, a long, slender body shape, chest deformities, scoliosis, and joint hypermobility [2]. These skeletal abnormalities, in combination with the frequent use of very thick eyeglasses, may result in a characteristic appearance that is different from the norm, and which is not only subjectively perceived by affected individuals but also visible to others. 

Research into the psychosocial consequences of Marfan syndrome is scarce [3]. Nevertheless, there is evidence that patients with Marfan syndrome are at risk for a number of adverse outcomes, including a decreased quality of life [4] particularly in the psychological domain [5]; challenges in education, work, and family life; depression and anxiety [6]; and fatigue [7,8]. However, to date, the question of whether the negative psychological consequences of Marfan syndrome extend to patients’ body image has not been addressed. The first goal of this study was to assess the extent of body image disturbances in Marfan patients. 

Body image can be conceptualized by means of two dimensions: (1) evaluation of the body and (2) perception of body dynamics [9]. Body evaluation comprises both the critical appraisal of one’s own appearance and the sense of well-being in one’s own body. Body dynamics refers to the perception of energy and vitality. This dimension describes the experience of power, fitness, and health related to one’s own body. 

So far, we do not know how these dimensions of body image are influenced by the interaction of psychological and physical factors in patients with chronic diseases in general and in patients with Marfan syndrome in particular. It was the second goal of this study to determine the extent of influence of potentially predictive factors on body image in individuals with Marfan syndrome.

Primarily three groups of potential predictors are considered in this study, namely (1) indicators of the physical peculiarities associated with Marfan syndrome (e.g., patients’ height and physical fitness), (2) variables associated with psychosocial health and distress, and (3) sociodemographic variables. 

### 1.1. Physical Factors

It is well known from research in developmental and social psychology that height influences perceptions of attractiveness and body image [10]. The skeletal abnormalities in Marfan syndrome result in a long, slender body shape with an above average body height. So far, it is unclear to what extent body image is affected by an individual being extremely tall. Exercise and physical fitness are known to have significant effects on body image [11]. Patients with Marfan syndrome often experience a lack of physical fitness. The degree of impairment and the experience of surgery are also validated risk factors for a negative body image in patients with congenital heart disease [12]. Therefore, height, physical fitness, degree of impairment, and a history of surgery can be expected to be predictors of body image in Marfan patients.

### 1.2. Psychosocial Factors

In terms of psychosocial factors, depression, anxiety, and somatization are known to be associated with body image. Significant associations between depression severity and body dissatisfaction can be found in adults [13,14,15,16,17]. Not only depression but also anxiety was positively correlated with body image disturbance in surgical patients with colorectal disease [18]. Furthermore, a link between anxiety and body dissatisfaction was shown for patients with anxiety disorders [19]. We also know that body image tends to be more negative in patients with somatoform or somatic symptom disorder compared with control subjects from the general population [20], indicating that somatization might also be an indicator of an impaired body image. 

### 1.3. Sociodemographic Factors

Finally, sociodemographic variables such as age [21], gender [22], and level of education as an indicator of socioeconomic status [23] may influence body image in general as well as in chronic diseases. Gender might be especially relevant because rates of body dissatisfaction are known to be higher in women than in men [24]. 

The goal of this study was to assess body image impairments in Marfan syndrome and to determine the extent of influence of potentially predictive factors (i.e., physical, psychosocial, and sociodemographic factors) on body image in individuals with Marfan syndrome.

## 2. Methods

### 2.1. Study Design

Using a cross-sectional study design we assessed 42 patients in terms of body image, physical condition, psychosocial health, and sociodemographic variables at the beginning of a rehabilitation treatment. We operationalized body image by means of a psychometric questionnaire on two dimensions, namely (1) Negative Body Evaluation and (2) Vital Body Dynamics, and compared the body image of Marfan patients with a German representative sample. Two separate backwards regression analyses were carried out to identify variables with the highest predictive value for the two dimensions of body image. Predictor variables comprised patients’ physical condition, psychosocial health, and sociodemographic variables. 

### 2.2. Patients

Assessment took place at the beginning of an inpatient rehabilitation program [25] in the Muehlenberg Clinic, a rehabilitation clinic in Germany specializing in cardiac rehabilitation. All patients met the following criteria: (1) diagnosis of Marfan syndrome (Q87.4 according to ICD-10- GM) in stable condition proven by a Marfan specialist located in one of the Marfan units in a German university clinic, (2) New York Heart Association (NYHA) stage < III, (3) time since last cardiovascular surgery > 3 months, and (4) no relevant increase in aortic diameter within the last 12 months. Exclusion criteria were severe psychiatric or other medical comorbidities. Five groups of patients comprising eight individuals recruited in 2014, ten individuals recruited in 2015, nine in 2016, ten in 2017, and eight in 2018, respectively, were included. Once patients were admitted to the rehabilitation program we offered participation in this study. One patient refused to participate. All the other 42 patients gave their informed consent to participate. 

### 2.3. Ethics Approval and Consent to Participate

All subjects were adults and gave their informed consent for inclusion before they participated in the study. The study was conducted in accordance with the Declaration of Helsinki and was approved by the ethics committee of the Medical Association of Hamburg (project identification code: PV4139).

### 2.4. Procedure

All patients answered a set of psychometric questionnaires on the second day of the rehabilitation program. Measures of physical fitness were recorded within the first 3 days after admission and documented in the medical records. 

## 3. Measurement Instruments

### 3.1. Body Image

All participants filled out a body image questionnaire (BIQ German: Fragebogen zum Körperbild, FKB-20) [9]. The BIQ is a short 20-item questionnaire which assesses body image disturbances on two scales: (1) Negative Body Evaluation and (2) Vital Body Dynamics. The first scale focuses on the subjective evaluation of one’s own body and on how a person feels within her or his own body (e.g., “Sometimes I wish to look completely different”). The second scale assesses the perception of energy, power, fitness, and health located in one’s own body (e.g., “I feel powerful”). With Cronbach’s alpha = 0.84 for both scales, reliability was found to be good. Validity has been demonstrated in previous studies by discriminating different clinical groups and clinical from nonclinical groups [9]. 

### 3.2. Physical Condition 

In order to assess physical fitness, we documented maximum distance for Nordic walking as well as maximum power in bicycle ergometry in the medical records for all patients. Maximum power in bicycle ergometry was defined as maximum Watts per kilogram bodyweight that patients were able to perform for at least 20 min with RR ≤ 160 mm Hg. We assessed fitness status in terms of these two parameters within the first 3 days after admission. In addition, we assessed NYHA Functional Class as an indicator of impairment and prior experiences of surgery to describe the patients’ physical condition. 

### 3.3. Psychosocial Health

In order to assess the psychological aspects of patients’ health status, patients filled out German versions of a series of questionnaires. We obtained sum scores for depression and anxiety using the Hospital Anxiety and Depression Scale (HADS) [26,27], a 14-item scale with seven items targeting anxiety and the other seven targeting depression. The HADS is a widely used instrument for the assessment of anxiety and depression in patients in medical treatment settings. Its reliability and validity are well established. 

We assessed somatization (i.e., suffering from diverse physical complaints) by means of the Somatization subscale of the Symptom Checklist-90-R (SCL-90-R) [28,29]. The SCL-90-R is a frequently used measure of psychological distress in clinical practice and research. A high number of studies have been conducted demonstrating the reliability, validity, and utility of this instrument. 

### 3.4. Sociodemographic Variables

Age, years of education as an indicator of socioeconomic status, and gender were extracted from patients’ medical records. Dependent variables and predictor variables are listed in Table 1.

### 3.5. Data Analyses

We calculated mean BIQ scores of Marfan patients separately for men and women and compared them with the German representative norming sample [30]. We carried out two separate regression analyses in order to determine which variables may predict the psychological and physical dimensions of patients’ body image (i.e., Negative Body Evaluation and Vital Body Dynamics, respectively). Physical fitness, NYHA Functional Class as an indicator of impairment, the experience of surgery, depression, anxiety, somatization, age, years of education as an indicator of socioeconomic status, and gender were entered into the regression as predictors. We used backwards regression, where only those variables that have a significant or marginally significant effect on the respective outcome remain in the final model. All analyses were performed using IBM SPSS Statistics, version 21. Unless otherwise specified, we expressed continuous data as means (m) ± standard deviation (sd) and categorical data as absolute numbers with respective percentages in parentheses. All tests were performed for exploratory means.

## 4. Results

Patients’ mean age was 45.1 years, 79% were female, 21% male. The educational level was relatively high, and most of the patients lived in stable relationships. Sixty percent had children. Further sociodemographic data are presented in Table 2.

All 42 individuals were diagnosed as Marfan patients according to the Ghent criteria [31]. Thirty-one patients had undergone operations of the aorta, the mitral valve, or of the popliteal artery (*n* = 1). One patient had undergone frequent eye surgery (*n* = 1). None of the operations had taken place immediately prior to the rehabilitation. According to the New York Heart Association (NYHA) Functional Classification, 76% of the patients (*n* = 32) had no cardiovascular symptoms and no limitation in regards to ordinary physical activity (NYHA I), whereas 24% (*n* = 10) showed mild symptoms and slight limitation during ordinary activity (NYHA II). Mean height was 182 ± 5.6 cm in female patients and 195 ± 6.0 cm in male patients, respectively. Patients’ individual height in centimeters was converted into percentiles by referencing to the German general population, stratified by sex [32]. The mean percentile of female patients was 95.9 ± 7.2. For male patients, the mean percentile was 95.1 ± 7.9. Table 3 presents cardiovascular manifestations and related previous surgical treatment.

Twelve patients were diagnosed with psychiatric comorbidities according to ICD-10, specifically major depressive episode (F32/F33) in four patients, agoraphobia (F40.0) in one patient, adjustment disorder (F43.2) in two patients, somatoform pain disorder (F45.41) in three patients, neurasthenia (F48.0) in one patient, anorexia nervosa (F50.1) in one patient, and insomnia (F51.0) in one patient. 

Mean scores for Vital Body Dynamics were 24.00 ± 6.522 for women and 23.11 ± 5.349 for men. Mean scores for Negative Body Evaluation were 28.53 ± 8.148 for women and 27.33 ± 9.014 for men. Compared with the German representative norming sample (*n* = 2473) of the BIQ (Albani et al., 2006) the body image of individuals with Marfan syndrome was significantly impaired, with *t* = −6.3 (*p* < 0.001) for Vital Body Dynamics and *t* = 7.6 (*p* < 0.001) for Negative Body Evaluation in women, and with *t* = −4.44 (*p* < 0.001) for Vital Body Dynamics and *t* = 4.31 (*p* < 0.001) for Negative Body Evaluation in men. The percentile rank of women with Marfan syndrome in their respective age cohort was 88 for Negative Body Evaluation and 11 for Vital Body Dynamics. For men with Marfan syndrome, the percentile rank was 91 for Negative Body Evaluation and 4 for Vital Body Dynamics.

For Vital Body Dynamics, the final regression model contained two predictors. Significant effects were observed for the variables age and depression, with older patients and patients scoring higher on the depression scale perceiving themselves as less vital. The final regression model on Negative Body Evaluation revealed a significant effect of anxiety, with higher scores predicting a more negative body image. No other effects reached significance. For regression coefficients and statistics, see Table 4. 

## 5. Discussion

This study set out to explore how body image in patients with Marfan syndrome might differ from the nonclinical population and which factors in particular might predict putative body image impairments. In line with our hypotheses, we found the body image of individuals with Marfan syndrome to be substantially impaired. Different factors predicted the two dimensions of body image, respectively. Out of a range of potentially predicting variables, anxiety best predicted Negative Body Evaluation, that is, critical appraisal of one’s own appearance and the sense of well-being in one’s own body, whereas a combination of age and depression best predicted Vital Body Dynamics, i.e., the perception of energy and vitality in one’s own body. 

### 5.1. Body Image of Patients with Marfan Syndrome Compared to the General Population

The first purpose of this study was to assess the extent of body image disturbances in Marfan patients. Comparing the body image of Marfan patients with the German representative norming sample of the BIQ [29], we found a significantly impaired body image in individuals with Marfan syndrome compared to the norming sample. The high percentile ranks for Negative Body Evaluation and the low percentile ranks for Vital Body Dynamics indicate that, in line with our hypotheses, Marfan syndrome patients do not feel well in their body, tend to evaluate their body much more negatively, and perceived themselves as much less powerful and energetic than healthy individuals of a comparable age.

### 5.2. Prediction of Body Image 

The second purpose of this study was to explore whether any sociodemographic variables and variables in the physical and the psychological domain might predict patients’ satisfaction with body image. The most important result in this respect is that except for patients’ age, mainly psychological variables predicted differences in body image in individuals with Marfan syndrome. Neither disease-related variables such as the experience of surgery or the degree of impairment indexed by the NYHA stage, nor variables indicating objective physical fitness such as maximum Nordic walking distance and maximum power in bicycle ergometry, had a more powerful impact on body image than the psychological state of health, more specifically, depression in combination with age, and anxiety. 

### 5.3. Vital Body Dynamics

For Vital Body Dynamics, patients’ age and their score on the depression scale had the highest predictive value. Understanding how body image functions and relates to psychological well-being across the life span in different groups has been subject to research by Midlarsky and Morin [21]. They found a decrease in the self-perception of vitality and attractiveness in the course of adult life not only in clinical samples but also in the general population. These findings are in line with the findings of Albani and colleagues [30] in the German general population showing a decrease of Vital Body Dynamics over the adult life span. As individuals age, the physical body undergoes many changes, and the physical body is not the same as it was in young adulthood. The loss of a youthful body may generally be difficult to accept, and the adaptation to the new physical realities of one’s body may be even more difficult in the presence of a congenital condition such as Marfan syndrome.

Feelings of depression contributed additionally to decreased Vital Body Dynamics, i.e., the perception of less vitality, energy, power, fitness, and health in regards to one’s own body. Similar results were reported by Paans and colleagues [33] who also found an independent contribution of depression to body image dissatisfaction in adults. These results are plausible given the fact that a lack of energy is one of the physical symptoms of depression. 

### 5.4. Negative Body Evaluation

For Negative Body Evaluation, i.e., both the critical appraisal of one’s own appearance and the sense of well-being in one’s own body, we found patients’ anxiety levels as assessed by the HADS to have the highest predictive value. A link between anxiety and body image has also been described for patients with other medical conditions including visible disfigurement [34] and surgery in colorectal disease [18]. Patients with higher levels of anxiety might feel more threatened by Marfan syndrome, and this may lead them to evaluate their own body more negatively. Furthermore, the above-average height that is typically observed in Marfan syndrome patients and that was also found in this sample might be associated with a negative body evaluation. Particularly in the case of social anxiety, the unusual appearance of individuals with Marfan syndrome might lead to feelings of insecurity in social interactions, which are then attributed to body image. A negative evaluation of the body follows congruously.

### 5.5. Study Limitations

The study design is cross-sectional, and the presented data are correlational. Correlation does not prove or imply causation, and this is true also for a stepwise regression model. Therefore, we cannot conclude that depression, age, and anxiety have a causal negative impact on body image. To justify stronger conclusions, longitudinal data are needed. However, the backward regression model is a reliable way of selecting the best grouping of predictor variables that account for the most variance in the outcome. 

With 42 participants, the sample size is rather small, limiting conclusive considerations. Including more patients would contribute to more valid results. However, given the rarity of Marfan syndrome, a sample size of 42 is considerable and provides at least some evidence.

Since this study is the first one to investigate body image in patients with Marfan syndrome and could not build on previous findings, it should be considered explorative. Nevertheless, evidence from studies with comparable populations informed the selection of variables and measures applied in this study.


Finally, other factors than the ones included in this study might contribute to understanding body image in patients with Marfan syndrome. One potential candidate might be family, as parents and siblings are powerful agents in shaping body image. Study participants might have had role models in dealing with the physical peculiarities and potential medical complications associated with Marfan syndrome in their families. Here, we did not have information about participants’ family history with Marfan syndrome. Future studies should take relevant aspects of family history into account. 

## 6. Conclusions

In the present study, we found that patients with Marfan syndrome evaluate their body much more negatively and feel less powerful and energetic than do individuals from the general population. This clearly indicates the need to address the issue of body image in future research efforts and in the treatment for Marfan syndrome. 

The results further indicate that body image in individuals with Marfan syndrome is not primarily predicted by objective measures of fitness or cardiac impairment. Instead, psychological variables like depression and anxiety determine the way individuals with Marfan syndrome experience their body. This finding is plausible because being visibly different, together with expected or perceived negative reactions from others, likely has an impact on psychological functioning and may especially result in a compromised body image. The finding that body image is mainly predicted by depression and anxiety can inform treatment and rehabilitation concepts. 

As Marfan patients nowadays tend to reach a higher age due to optimized medical and surgical management, it may be important to pay special attention to older patients. In particular, it could be beneficial to monitor depression and anxiety in this group by means of screening questionnaires and, if necessary, provide treatment in order to support patients in coping with and accepting the ramifications of Marfan syndrome. If body image satisfaction and disease acceptance are a goal, therapy and rehabilitation should not only include medical treatment and aspects of physical rehabilitation, but should also comprise psychological interventions targeted at depression and anxiety.

## Figures and Tables

**Table 1 jcm-09-01015-t001:** Dependent and predictor variables.

Dependent Variables
Vital Body Dynamics
Negative Body Evaluations
**Predictor Variables**
Physical condition
Physical fitness (W/kg body weight in bicycle ergometry)
NYHA Functional Class
Experience of surgery
Psychosocial health
Depression
Anxiety
Somatization
Sociodemographic variables
Age
Years of education
Gender

**Table 2 jcm-09-01015-t002:** Sociodemographic data.

Variable	
Age	m ± sd 45.1 ± 8.3
Years of school	m ± sd
education	11 ± 1.6
Gender	n (%)
Female	33 (79)
Male	9 (21)
Professional qualification	n (%)
Blue collar	7 (17)
White collar	35 (83)
Marital status	n (%)
Single/no partner	8 (19)
Married	29 (69)
Divorced/separated	5 (12)
Children	n (%)
0	17 (40)
1	7 (17)
2	17 (40)
3	1 (2)

m: mean; sd: standard deviation; n: number of patients.

**Table 3 jcm-09-01015-t003:** Cardiovascular manifestations and surgical/pharmacological treatment.

Cardiovascular Manifestations Previously Treated by Surgery	Number of Patients Affected
Mitral valve prolapse	4
Tricuspid valve prolapse	1
Aortic dissection Stanford A	2
Aortic dissection Stanford B	5
Aortic regurgitation	13
Aortic aneurysm	18
Mitral regurgitation	3
Patent foramen ovale	1
Pulmonary artery dilatation	1
**Cardiovascular Manifestations Not Yet Treated By Surgery**	
Mild mitral valve prolapse	4
Ascending aorta < 45 mm	5
**Previous Cardiovascular Surgery**	**Number of Patients Treated With**
David procedure	15
TEVAR procedure	1
Bentall procedure	1
Aortic valve replacement	9
Mitral valve replacement	1
Aortic prosthesis	10
Aortic valve reconstruction	6
Mitral valve reconstruction	3
Aneurysmectomy popliteal artery	1
Aortic root reconstruction	1
Patent foramen ovale closure	1

**Table 4 jcm-09-01015-t004:** Unstandardized regression coefficients (b), t-values, and significance values (p) of the regression analyses for Vital Body Dynamics and Negative Body Evaluations.

Dependent Variable	Predictor	*b*	*t*	*p*
Vital Body Dynamics	Age	−0.218	−2.518	0.016
	Depression	−0.834	−5.119	<0.001
Negative Body Evaluations	Anxiety	0.967	3.759	0.001

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
