# Peer review of "Body Image in Patients with Marfan Syndrome"

_jcm, 2020, doi:10.3390/jcm9041015_

Round 1

Reviewer 1 Report

I read your work with great interest, because I found it interesting that researchers also begin to treat the quality of life of patients with Marfan, evaluating all the factors that can influence his clinical status, disease progression but not only but also psychophysical and psychosocial. Having a general picture of the state of all these variables and showing their correlations, I found it interesting ...
The data that I found brilliant are the correlations of the predictor, age (demographic), depression (psychophysical) and anxiety (psychosocial) variables with Vital Body Dynamics and Negative Body Evaluations. In female patients they seem more accentuated .. certainly the small sample size limits conclusive considerations and leads to suggest to widen it, as well as to perform a longitudinal study

Author Response

Author's Reply to the Review Report (Reviewer 1)

We appreciate greatly the reviewer’s encouraging and supportive comments to our study.

We incorporated two remarks in the text.

Point 1:

In female patients they seem more accentuated

Point 2:

certainly the small sample size limits conclusive considerations

We concur with reviewer 1 in that the small sample size limits conclusive considerations and that longitudinal data would be desirable. These aspects are emphasized in the discussion (lines 265-266, and line 268, respectively).

As per suggestion of reviewer 1 who requested moderate English changed, the final version of the manuscript has undergone an additional round of proofreading and spellchecking by a native speaker of English.

Reviewer 2 Report

Overall, this is an interesting paper. Analyses are performed adequately and limitations are well acknowledged in this paper.  The only suggestion I would like to make is to take family history into account. In my experience, the fact that family members have been suffering from complications from marfan syndrome does affect patients (self)perception. If not available, this should be mentioned in the limitations 

Author Response

We are grateful for the reviewer’s positive remarks and for the helpful note regarding patients’ family history with Marfan syndrome.

Point 1:

The only suggestion I would like to make is to take family history into account. In my experience, the fact that family members have been suffering from complications from marfan syndrome does affect patients (self)perception. If not available, this should be mentioned in the limitations

As suggested by reviewer 2, we now acknowledge in the section on limitations that we do not have any information regarding the family history of our patients (see lines 275-281). We agree with the reviewer that having seen family members suffer from complications from Marfan syndrome may very well affect patients’ self-perception. Thank you for this insightful comment.

Reviewer 3 Report

The present manuscript by Hansen et al reports on the body image impairments of patients with Marfan syndrome. The authors conclude that body image is mainly determined by psychological variables and age, rather than objective measures of fitness or cardiac impairment. The findings are insightful and provide an appreciation of changes in psychological factors in this patient population. However, there are some major flaws and concerns in the study that warrant urgent attention from the authors.

Comments to the Author

  1. Lines 59-84 (page 2), the authors should move all the background information to the Introduction.
  2. Lines 97-99 (page 3), the authors indicate that patients with diagnosis of Marfan syndrome or a similar syndrome were included in the study. Can they clarify what they mean by “similar syndrome?” I believe the analysis should be limited to a population with only Marfan syndrome, not a heterogenous cohort.
  3. In the Data analyses, which software did the authors use for their analysis? There should be a sentence or two to describe how the data were presented in the manuscript. Additionally, it is very concerning that the authors used an already published cohort as the control population. Can they comment on this?
  4. Lines 165-177 (page 5), the authors state that only 42 patients were diagnosed as having Marfan syndrome, with the two others having Loeys-Dietz- and Ehlers-Danlos-syndromes. I am concerned that these two diseases were not excluded during the patient recruitment step.
  5. The presentation of means and standard deviations as mean=xxx (sd=xxx) should be corrected. The standard way is to have mean±SD.
  6. In the Discussion (page 6, lines 202-208), the authors repeated the reporting of results, which is clearly not in accordance with standard scientific writing. Herein, the authors should summarise the main findings and discuss them in subsequent paragraphs.

Author Response

We thank the reviewer for the careful reading of the manuscript and the associated helpful and important feedback provided.

Point 1:

Lines 59-84 (page 2), the authors should move all the background information to the Introduction.

The introduction was restructured by moving the considerations regarding potential predictors of body image to the background section of the manuscript (lines 58-85).

Point 2 and 4:

Lines 97-99 (page 3), the authors indicate that patients with diagnosis of Marfan syndrome or a similar syndrome were included in the study. Can they clarify what they mean by “similar syndrome?” I believe the analysis should be limited to a population with only Marfan syndrome, not a heterogenous cohort.

Lines 165-177 (page 5), the authors state that only 42 patients were diagnosed as having Marfan syndrome, with the two others having Loeys-Dietz- and Ehlers-Danlos-syndromes. I am concerned that these two diseases were not excluded during the patient recruitment step.

All analyses are now limited to patients with Marfan syndrome (see the section on “Patients”, lines 97-107). The two individuals with Ehlers-Danlos- and Loeys-Dietz-syndrome are excluded from the study. All reported results are corrected accordingly. The exclusion of the two patients with a syndrome similar to Marfan did not affect the outcome substantially. Thank you for this helpful suggestion, we agree with your concern.

Point 3:

In the Data analyses, which software did the authors use for their analysis?

The statistical program used is now mentioned in the methods section (lines 159-160).

There should be a sentence or two to describe how the data were presented in the manuscript.

We write in our “Methods”, section “Data analyses”: „Unless otherwise specified, we expressed continuous data as means ± standard deviation and categorical data as absolute numbers with respective percentages in parentheses. All tests were performed for exploratory means”. (see lines 160-162)

Additionally, it is very concerning that the authors used an already published cohort as the control population. Can they comment on this.

Our aim was to verify body image disturbances in our clinical sample as a necessary antecedent to assessing which factors might predict these disturbances. For this purpose, we found a comparison with a large, normative sample to more useful and informative than testing against a smaller-scale experimental control group. For this reason, we report both percentiles and tests results. We thank the reviewer for this remark.

Point 5:

The presentation of means and standard deviations as mean=xxx (sd=xxx) should be corrected. The standard way is to have mean±SD.

We changed the presentation of means and standard deviations in the text and the tables to “mean±sd”. We thank the reviewer for bringing it to our attention.

Point 6:

In the Discussion (page 6, lines 202-208), the authors repeated the reporting of results, which is clearly not in accordance with standard scientific writing. Herein, the authors should summarise the main findings and discuss them in subsequent paragraphs.

The first paragraph of the discussion is rewritten as it now summarizes the main findings without again reporting results (lines 206-208), as suggested by reviewer 3. Thank you for bringing up these points!